# How to Build Live-Cell Sensor Microdevices

**DOI:** 10.3390/s23083886

**Published:** 2023-04-11

**Authors:** Pelagia-Irene Gouma

**Affiliations:** Department of Materials Science and Engineering, The Ohio State University, Columbus, OH 43210, USA; gouma.2@osu.edu; Tel.: +1-614-292-4931

**Keywords:** live cell sensor, scaffold, volatolomics, pathogen detection, environmental monitoring

## Abstract

There is a lot of discussion on how viruses (such as influenza and SARS-CoV-2) are transmitted in air, potentially from aerosols and respiratory droplets, and thus it is important to monitor the environment for the presence of an active pathogen. Currently, the presence of viruses is being determined using primarily nucleic acid-based detection methods, such as reverse transcription- polymerase chain reaction (RT-PCR) tests. Antigen tests have also been developed for this purpose. However, most nucleic acid and antigen methods fail to discriminate between a viable and a non-viable virus. Therefore, we present an alternative, innovative, and disruptive approach involving a live-cell sensor microdevice that captures the viruses (and bacteria) from the air, becomes infected by them, and emits signals for an early warning of the presence of pathogens. This perspective outlines the processes and components required for living sensors to monitor the presence of pathogens in built environments and highlights the opportunity to use immune sentinels in the cells of normal human skin to produce monitors for indoor air pollutants.

## 1. How to Design Living Cell Sensors

Viruses are transmitted in air and thus monitoring the presence of airborne pathogens is important, as is the need to find better ways to detect and measure infectious viruses in the air [1]. The design, modeling, fabrication, and manufacturing of engineered living systems as components of devices capable of performing tasks, such as environmental responsiveness, were addressed in this perspective. There is a variety of transduction mechanisms that can be explored, including electrical, optical, and mechanical outputs. In previous studies [2,3], sensor arrays for environmental and health monitoring were demonstrated, along with the potential of the electrospinning process to produce bio-composite mats of water-soluble polymers with live biomolecules (enzymes) which were subsequently used as receptors for urea-sensing with a shelf-life of several months. Three-dimensional hierarchical scaffolds mimicking the extra-cellular matrix (ECM) of the porcine urinary bladder have also been produced [4]. The author, as well as other workers in the field, has produced fundamental knowledge of the electrostatic drawing process and demonstrated novel electrospinning tools. Complex, self-supported, 3D nanostructured sensors and photocatalytic materials (nanogrids) and one-of-a-kind computational models for mapping fibrous architectures of functional materials have become available (see citations in [3]). These fibrous scaffolds for cell growth can be utilized to grow cell cultures that give a chemical signal in the presence of a pathogen according to volatolomics [2,5]. A scalable high-throughput electrospinning process has been demonstrated by the author and will presented later in this perspective [3]. Three-dimensional printed electrospun scaffolds used in tissue and bone engineering have also been demonstrated, as presented below, to promote the growth of 3D cell cultures.

Based on the host response of a subject—in this case the gases emitted from cells in the case of infection by a virus—a novel sensor system/breath tester for COVID-19 which produced a breath print for the infection by “smelling” a single exhaled breath was also demonstrated [6]. At the same time, there are chemo-mechanical actuators and resistors for sensing gases emitted from skin cells [7]. These breakthroughs have opened the way to explore fundamentally novel approaches to safeguard the environment, protect public health and individual well-being through the processing of living sensor systems that detect pathogens rapidly, as described next. 

This perspective focuses primarily on electrospinning suspensions of microorganisms and polymers to produce living fibers and the use of dermal sentinels of the immune system to obtain a rapid response to infections. It also addresses the ethical and social issues in bioengineering that are pertaining to the use of live cells as sensor systems. Figure 1 refers to the main themes of this work.

The key technical challenge is the response time from capturing the virus to infecting the cell culture and obtaining a clear signal of infection. It is important to have a release of the gases signaling infection (that is live virus) within a reasonable timeframe to prevent the virus spreading to humans. Thus, the focus should be on rapidly interfacing the membrane filters (i.e., non-woven fabrics that capture the viruses during air circulation) with the 3D cultured cell scaffolds sampling both the headspace and the supernatant for these gaseous biomarkers of infection. The device should sample air through an inlet (interfaced with a flow controller). The cassette with the cell-culture microdevice is expected to be replaced every two weeks or longer. This is an original and unique approach to detecting viable virus particles in air and is amenable to be modified to provide information on many different viruses based on which biomarkers are being detected. Ultimately, a library of signals should become available that will permit virus recognition for many different pathogens.

The innovative technologies highlighted here include: 3D nanofibrous scaffolds based on cellulose acetate to maintain cell viability for at least two weeks at a time;cell-scaffold formation by direct cell electrospinning;super-water-repellant filtration membranes;volatolomics, gaseous biomarkers, and novel sensors.

### 1.1. Cell Scaffolding 

Cell scaffolding typically aims to restore, maintain, or improve tissue function [8]. This occurs in nature using the 3D-structure of the ECM—a natural scaffold that allows cells to grow, proliferate and differentiate within it [9]. The ECM is a complex, three-dimensional ultrastructure of proteins, proteoglycans and glycoproteins, used to promote cell growth in native tissue [9]. In fact, there are many different types of ECMs for different parts of the body. For example, fibrous proteins are a dominant material in tendons, while polysaccharides are found to largely exist in cartilage. The ECM provides attachment sites and mechanical support for cells. In addition to the bioactive functional motifs present, the topology of ECM has also been found to affect the structure, functionality, and physiological responsiveness of cells. The geometry of the natural matrix was reported to modulate the cell polarity. Thyroid cells, smooth muscle cells and hepatocytes are different types of cells found to be affected by ECM’s topology, with 3D structures inducing cell differentiation more effectively than 2D configurations. 

The complex ECM structure involves multiple length scales, layers, and morphologies. The ECM isolated from the porcine small intestine or urinary bladder was shown to provide an environment suitable for regenerating many different types of tissue, including small-diameter arterial grafts, vena cava, and urinary bladder [10], all suggesting its potential as a scaffold capable of tissue remodeling. Still, there are limitations in the reproducibility of the topology of the material, sterilization, etc. 3D-printing technologies for the fabrication of scaffolds with complex architectures allows for the reproducible replication of native scaffold characteristics. The vasculature, cell–cell interactions, and the mechanical properties vary with the tissue type and sometimes within a certain tissue type. The ability to print multilayer configurations is important for the mimicking of heterogeneous tissues, such as musculoskeletal and dental tissues. In all, the flexible ability to print different scaffold types as bio-mimicking implants is important. Morphological features, such as the scaffold’s fiber size, porosity, and whether the scaffold is in compression or tension, all affect cell–scaffold interactions and are known to provide mechanical cues that drive cell attachment, growth, and even differentiation [8,9,10,11,12]. 

The common 3D-printing techniques were summarized in a recent review [8]: “extrusion printing/bioprinting, stereolithography (SLA), powder-fusion printing (PFP), laser-assisted bioprinting (LAB), and inkjet bioprinting”. Among these technologies, there are several limitations, such as high-temperature printing that is not compatible with biological molecules (cells, proteins, etc.), or the inability to support many layers, or the produced scaffolds having a limited height, or the process being very expensive [8]. Finally, cell electrospinning involves producing viable threads containing living cells [13]. While direct-cell electrospinning typically utilizes coaxial needles (core-shell electrospinning process) that do not make for a scalable process, recent efforts have focused on optimizing the strength of the field and the needle to the collector distance to produce hybrid fibers from a single jet [14].

Cellulose acetate (CA) thin, porous membranes were produced by electrospinning precursor polymer solutions in acetone at room temperature. These membranes were used as scaffolds for 3D-microvascular cell growth [11]. Human umbilical vein endothelial cells were obtained as first-passage cultures. Cells were incubated at 37 °C in a 5% CO_2_ humidified atmosphere. The electrospun materials were tested for their effect on cellular viability [11]. Furthermore, artificial bone-tissue scaffolds based on natural hybrids of cellulose acetate (CA) and nano-hydroxyapatite (n-HA) were also fabricated in a bio-mimicking 3D-matrix architecture using a single-step nanomanufacturing technique and were used for in vitro bone regeneration studies for up to 14 days [12]. Figure 2 shows 3D-cell cultures grown on ECM-mimicking matrices [4,11]. 

A simpler process involves directly electrospinning a solution of CA/PVP and cells on microfabricated silicon devices—the organ-on-chip design discussed in the reference. A serum-free liquid needs to be used as the buffer to keep the cell cultures active inside the fibrous mats. Following electrospinning, the chip will be packaged in a cassette made of a gas-proof polymer which will have two openings—one for adding water to dissolve PVP and liberate the cells and another for sampling the headspace for the gases released. The cassettes will be the consumables of the final device.

### 1.2. Processing of the Filtration Membranes

In the electrospinning process, the build-up of electrostatic charges on the surface of a liquid droplet induces the formation of a jet. The jet is subsequently stretched to form a continuous fiber. Before it reaches the collecting screen, the solvent evaporates or solidifies. The fibers form non-woven mats that are characterized by high surface areas and relatively small pore sizes. This improves the adsorption properties of the material. The simple recipe to follow requires the following precursors: cellulose acetate powders with the molecular mass of Mn~30,000 and Acetic acid and acetone with a volume ratio of solute:solution of 2:3. Electrospinning can be carried out using a high voltage of 20 kV at a flow rate of 30 μL/min and a working distance of 7 cm. These conditions will reproduce the morphology of the membrane/mats illustrated in Figure 3 from reference [15]. A process map is available for electrospun CA [4], a guide on modifying the structure of these filters as needed.

The physics of filtration is described in reference [16]. Particles larger than 50 nm can contact the fibers of the filter through the processes of interception and/or diffusion. Each SARSCoV-2 virion, for example, is approximately 50–200 nanometers in diameter. To capture these particles using a hydrophobic fibrous membrane, one needs to consider that particles smaller than 1 mm can adhere to the filter material (impaction) [16] as seen in Figure 4 [14]. Studies by other workers comparing hydrophobic with electrostatic breathing filters for virus and bacteria filtration have identified the former to perform better overall [17], thus illustrating the high merit of this approach. 

When the cell cultures are exposed to the virus (by either direct or aerosolized contact) they are expected to undergo quantifiable changes (morphological and biochemical) over time, as discussed in the following section. 

## 2. Cell Cultures as Living Sensors

A recent publication revealed that a three-dimensional alveolar-like (self-organized) stem-cell culture demonstrated an infection response to SARS-COV-2 [18]. Those authors used viral particles collected from a COVID-19 positive patient [18]. The infection triggered an innate immune response, even from a singular viral entry [18]. Given that the immune response was noticed in a very short time, it is feasible to prepare cell-culture models that can replicate this immune response upon infection with SARS-COV-2 and other viral and bacterial pathogens. Once the cell-response model is established, cells can then be exposed to actual or surrogates of human aerosolized pathogens. Gas-release levels in the headspace of the cell cultures may be sampled for the presence and concentration of disease biomarkers, such as NO and volatile organic compounds (VOCs), over time. Recently, we identified the olfactory pattern for COVID-19 infection and a breath test was developed for monitoring the pandemic disease [6]. This distinct breath print may be used to explore novel diagnostics for the environmental monitoring of pathogens based on diverse manifestations of the respective immune responses, including olfactory ones (*volatolomics approach*). Furthermore, certain components of skin cells (dermal DDCs and macrophages) may be explored for sensing pollutants in the environment, as they give immune response to irritants and inflammation [19]. The ability of dermal sentinels of the immune system to elicit a response to various infections may be measured and correlated to levels of gaseous disease biomarkers, such as NO and VOCs, over time. 

### 2.1. Volatolomics-Headspace and In-Vivo Chemo-Sensing of Nitric Oxide and VOCs Released from Cells

The sensing of volatile organic compounds emitted by cells in response to various perturbations (in this case onset of infection) involves the novel field of olfactory sciences namely, volatolomics [5]. This involves sampling the headspace of cells for signaling gaseous compounds which adds bio-chemical information to morphological variations (i.e., biomarker identification). Different VOC signatures for different types of cancerous cell lines using headspace sampling and an electronic nose-based sensor system have been detected [2]. Semiconducting oxides exhibit a strong affinity to specific compounds as a function of their polymorphic structure or crystallographic arrangement [2]. The selectivity of a sensor to a specific gas, or a class of gases, correlates with the metal-oxide polymorph used for sensing, and it is independent of the processing method used to produce the sensing material [20]. Because there is no control over the exposed crystal faces and planes in polycrystalline materials, there is limited gas selectivity and much cross-interference from other gases can be observed [21]. Furthermore, novel instrumentation, such as liquid-phase NO sensors, can also be considered as they may allow for in vivo sampling of individual cells for this specific biomarker. Novel semiconducting nanoprobes based on conducting polymers have also been developed for biomarker detection (see citations in ref [22,23]).

#### 2.1.1. Cell Culture Headspace Sensing

We may use the Formulas (1)–(3) shown below to identify headspace concentrations for a range of VOCs, to simulate the headspace above the cell cultures for the lab tests, and also utilize a gas-flow bench for probing the sensitivity limits of the sensors to the targeted gases. A water–air partition ratio of 300 is used for acetone calculations; 1300 is used for ethanol; 2600 and 3100 were used for calculations for 1-butanol, and 2-propanol, respectively. Because there are no widely agreed upon values for partition ratios of these chemicals, these values are an approximate average from several sources.
(1)$$Headspace\;Concentration\;=\;\frac{Concentration\;of\;Solution}{Phase\;Ratio\;+\;Water\;-\;Air\;Partition\;Ratio}$$
(2)$$Phase\;Ratio\;=\;\frac{Volume\;of\;Headspace}{Volume\;of\;Sample}$$
(3)$$Concentration\;of\;Solution\;=\;\frac{Original\;Weight\;of\;Chemical}{Volume\;of\;Sample}$$

#### 2.1.2. 3D Hybrid Scaffolds and Pathogen Signatures

Cell electrospinning has been demonstrated successfully as have organs on chip configurations. Modeling the direct electrospinning of live cells in fibrous scaffolds and design-optimized mat architectures for tailored efficiency in capturing the pathogens in air is required. It is also possible to tune the transduction modes to obtain a visible signal (color or bending). The novel nanotechnology tools described above are expected to offer highly sensitive and selective detection of pathogens through the output signal of the live cell-biosensors. Several novel transduction mechanisms may also be explored, such as optical and chemo-mechanical mechanisms. 

The sensing of volatile organic compounds emitted by cells in response to the onset of infection requires sampling the gases emitted by cells for these compounds. The cells’ culture sensing systems envisioned here shall augment the concept of “smart cell-cultures” [24,25,26,27] to produce ubiquitous and solid-state pathogen monitors. For this, “*tissue chips*” designs [27] may be implemented which will also incorporate semiconducting oxide nanowires or foams (see Figure 5 below). Novel monolithic single-crystal foams of metal oxides produced by electrospinning can also be explored as cell hosts [28]. Novel instrumentation (liquid-phase NO sensors) will allow in vivo sampling of individual cells for signatures of infections. 

## 3. Scalable Manufacturing and Validation of Air-Bio-Detectors

Virions are approximately 50–200 nanometers in diameter and Bacteria 1–2 mm. To capture pathogens, one needs to consider that particles smaller than 2 mm can adhere to the filter material (i.e., *induce impacted particle paths*). Among the various high-throughput electrospinning systems reviewed in ref [3] are the innovative spinneret and collector designs depicted in Figure 6 below (see ref [3] for details), which can achieve hybrid mats in a single-step process, depositing them on microfabricated silicon devices (tissue chips) for fully integrated microdevices, or producing aligned, self-supported, sensor-array configurations based on the preferred transduction mechanism and operating environment. Cellulose acetate-based scaffolds/filters which can be interfaced with the living sensors using the scalable electrospinning process are shown in Figure 6 below.

### 3.1. Ethics/Social Considerations

Such studies need to abide by the International Biosafety Protocols, and it is important to carry out a risk-to-benefit ratio analysis before deploying any of the living sensor technology products described here. Research products should produce infrastructure that will benefit mankind. The air-quality bio-detector devices described here may be used to sample the air in patients’ rooms in hospitals for pathogen detection. These need to run for a few minutes every hour over the course of two weeks to allow proper validation of the technology. 

### 3.2. Transformative Impact

The innovative technologies discussed in this perspective include: 3D nanofibrous scaffolds based on cellulose acetate which both capture the viruses/bacteria and expose then directly to the live cells on them deposited via direct cell electrospinning, and the fabrication of self-standing live-cell devices that maintain the cell viability for considerable time; (ii) self- supported live-cell sensor array microsystems; and (iii) variable transducing modes for the display of the output signals (color-changing photonic assemblies; electronic displays; chemo-actuators, etc.). The live-cell devices shall sample ambient air. This is an original and unique approach to detecting viable virus particles in air and is amenable to be modified to provide information on many different viruses based on the activation of cell sentinels or the release and detection of specific gaseous and liquid biomarkers of infection and disease. Ultimately, a library of signals will permit virus recognition for many different pathogens. Such systems shall provide unique insights into the type and concentration of gaseous metabolic products of cells. They will revolutionize how we perceive and detect pathogens. 

State of the Art and Outlook: The recent pandemic inspired a lot of research activity in indoor air-sampling surveillance for the presence of pathogens. Almost all of them involved RNA extraction from the virus [29]. In one study air samples were collected together with swab samples every eight hours [30]. It was found that air sampling is a more reliable tool for detecting the pathogens than the swabbing of surfaces. A UK-based company (Kromek) has presented a medium refrigerator-sized device using a flow rate of 400 lpm and condensing any biological materials in air into a single droplet of water, then employing next-generation sequencing processes to read and compare the genome of viruses against a database of existing pathogens stored within the system [31]. The entire identification process is expected to take place at the point of collection, and it takes around 30 min. Of course, this is a costly and timely solution, and it is not clear what the detection threshold is for this technology that is currently under development—their biosequencer platform is currently at Technology Readiness Level (TRL) 6 [32]. There are other biosensors being developed, such as aptamer-functionalized transistors, which can measure trace-level liquid samples (0.3 μL) and even gaseous-media samples at an ultra-low concentration (0.1 fg/mL) [33]. These employ ion-gated transistors with multi-channel analysis can respond to multiple targets simultaneously within as little as 10 min. These still require a significant amount of virus particles to be present and their response is quite long. Finally, Smiths Detection has launched a Bioflash Biological Identifier [34] based on the cellular analysis and notification of antigen risks and yields (CANARY) technology using genetically engineered immune cells that bind to a specific pathogen. There is not sufficient public information regarding the performance of this device.

Live-Cell Sensor Microdevices can become a gamechanger as they will identify when and where a clear danger from a viable virus exists, as opposed to the risk of spreading panic over the presence of dead virus segments, as can be the case when using solely nucleic acid-based technologies. These novel sensors will respond rapidly, thus enabling constant environmental monitoring. They will adapt engineering concepts to natural responses to infection, thus becoming novel biosensors and health diagnostics. This is a novel merging of biology and engineering, where we will use nature’s immune response to pathogens to inform us of the presence of the latter in the environment. Cells will be treated as (biological) engineering materials processed by manufacturing techniques in assemblies/arrays, living organisms producing immune responses to viral or microbial invasions which will be captured and transduced using new modalities for easy and early warnings. The anticipated impact of such a sensor technology is expected to be totally transformative, so it may lead to new hybrid materials and environmental sentinel species, as well as advancing engineering practices for the creation of a workforce ready to apply the new knowledge to construct innovative products. 

## Figures and Tables

**Figure 1 sensors-23-03886-f001:**
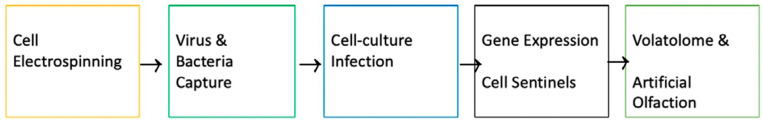
The main themes addressed in this perspective span across disciplines from sensors to immunology and from nano-bio-manufacturing to ethics.

**Figure 2 sensors-23-03886-f002:**
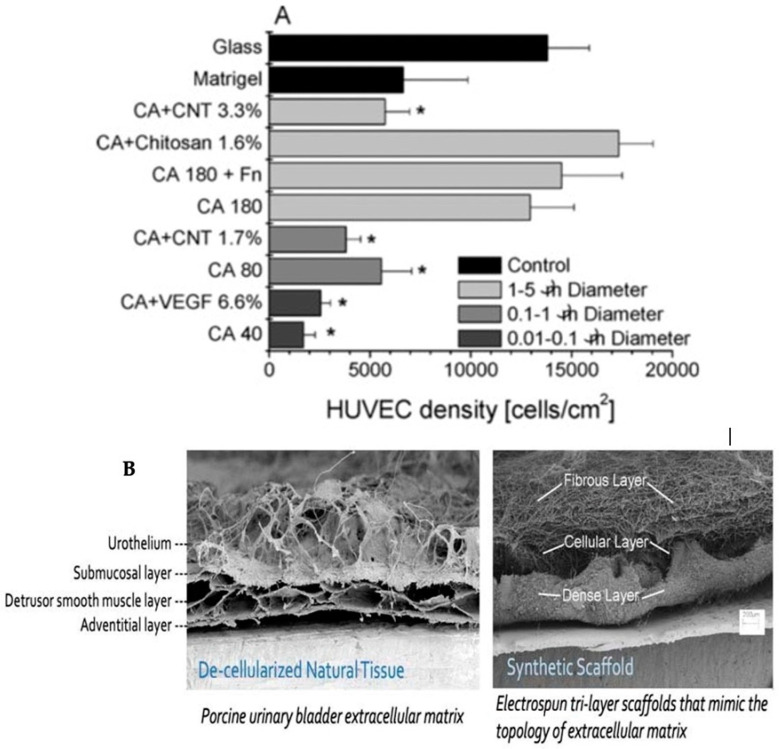
(**A**) The cell density on the glass coverslip, or within the scaffold; * cell density differs from on glass [11]; (**B**) left: the natural scaffold (ECM); right: synthetic scaffold [4].

**Figure 3 sensors-23-03886-f003:**
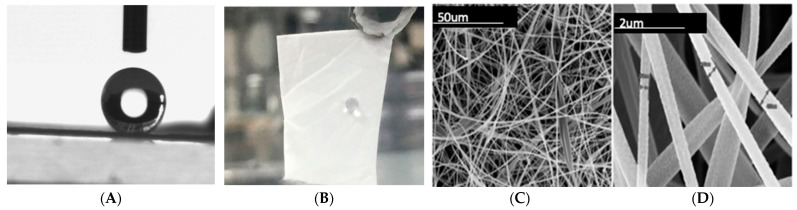
Super-water-repellant cellulose acetate electrospun mats for filtration applications [15]: (**A**) water contact angle on CA mat; (**B**) super-water-repellant nature of the CA mat; (**C**) low-magnification SEM image of the fibrous mat; and (**D**) high-magnification SEM image showing the fiber characteristics.

**Figure 4 sensors-23-03886-f004:**
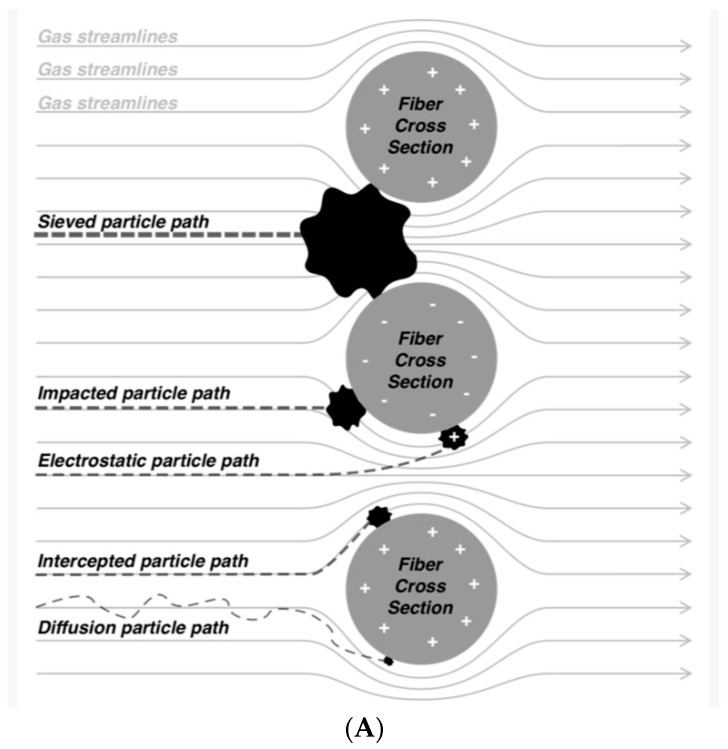
(**A**) Various phenomena involved in the filtration process using a fibrous membrane; (**B**) (virus) particle size vs. filtration efficiency (from [16]).

**Figure 5 sensors-23-03886-f005:**
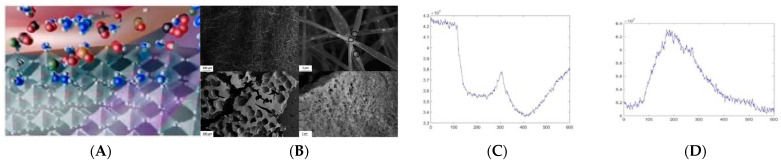
(**A**) Illustration of selective ammonia (blue molecules) chemosensing by metal oxide (MoO_3_) nanoprobes; (**B**) metal-oxide (WO_3_) foam structures [28]; and (**C**) and (**D**) output signals from sensing volatile biomarkers for pathogen vs. normal response, respectively [6].

**Figure 6 sensors-23-03886-f006:**
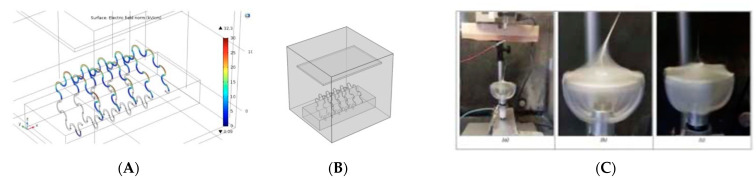
(**A**) COMSOL design of a novel spinneret and its characteristic electrostatic field; (**B**) schematic of the novel electrospinning setup; and (**C**) novel rotating collector that permits multiple fiber patterns—such as aligned vs. random—offering design choices for the living sensor arrays [3].

## Data Availability

Data sharing not applicable.

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
