# Peer review of "How to Build Live-Cell Sensor Microdevices"

_sensors, 2023, doi:10.3390/s23083886_

Round 1

Reviewer 1 Report

The present paper comprises an approach involving a live cell sensor microdevice that captures the viruses/bacteria from the environment and emits a signals to allow an early warning of the presence of pathogens. This perspective summaries relevant information on the processes and components required for living sensors to monitor the presence of pathogens in built environments.

However, this review requires some minor revisions, namely:

I) The Figures quality must be improved in all of them. Also, the numbers and letters are very small.

II) The (i) is missing in line 260.

III) References need to be formatted.

Author Response

Dear Sir/Madam,

thank you for your thoughtful review.

We have indeed updated the editing of the manuscript as suggested, while maintaining the format given by the Journal.

Yours,

P. Gouma

Reviewer 2 Report

The Dr. Pelagia-Irene Gouma presented for the assessment of suitability for printing in the journal "Sensors" a work with the type of Perspective entitled How to Build Live-Cell Sensor Microdevices”, which can be published after enriching it with the state of the art, even in related fields, and removing a few minor errors discussed and/or listed below:

Opinion: Announcing in print the author's publishing plans on such an important topic may not be reasonable, due to the transfer of knowledge to competing laboratories, both scientific and industrial.

Comment: Fifteen self-citations by the author of Gouma P.I or Gouma P. out of a total of twenty-seven cited papers is definitely too much, because it is not about a review of the author's achievements compared to the state of knowledge, but the author's predictions in the direction of research in accordance with the title of the manuscript based on the author's state of knowledge. Please change the type of article, or better alternative, enrich the description with the actual state of the art. In conclusion, much more source literature by other authors is strongly desired before the article type of "Perspective" publication.

Minor remarks and comments:

At line 149 is: … 1um … . Check and correct um, probably to µm. Similar errors to be corrected are in lines 237 and 238.  

At line 185 is: … Earlier unpublished work by the author has identified … , however, please refer to the results of unpublished papers only after they have been accepted for publication or announced as a preprint, or alternatively mention your own studies. Then please provide a citation for the preprint.

At line 195 is: … material[ 18]. … , but should be … material [18]. … .

At line 212 is: ... 2.2.3. D hybrid scaffolds and Pathogen Signatures … , but please correct the subsection number probably to 2.1.2. .

At line 209 is: … these values are an approximate average from several sources. … , however, please provide these sources, it is necessary. Comment: Prospects should be based on the author's solid, well-documented knowledge, otherwise they become mere predictions about future research.

At line 213 is: … The author was the first … , but authors are usually the first to publish their findings in prestigious journals, so this text is completely redundant. Please correct the text. Comment: By the way, there were three equal authors, not one. Rather, the plural should have been used.

At lines 216–218 the reason for the italic font is not obvious? Comment: Please check and correct if necessary.

For example, line 237 ends with ... 1-2 ... , line 226 is ... 23-25 ... and line 303 says ... 37-41 ... but it should rather be ... 1-2 ... , ... 23-25 ... , and ... 37-41 ... , respectively, as it is on line 147 and in the middle of line 237 ( … 50–200 … ). Lately, the middle character " – " has been used more than the short character " - " used in typewriters in the past. Comment: Please correct throughout the manuscript including the literature cited paragraph.

Author Response

Dear Sir/Madam,

thank you for your thoughtful comments and especially for expressing your opinion why to share know how and ideas on this manuscript. I will answer to your comment first then. When the pandemic happened, there was no way to know who will get infected, where and how. Yes, there were swabs and rapid tests but these could prove someone is positive for days after the infection was gone and they would fail most of the infectious people in the early stages of the disease. Other ways to monitor the spreading of the virus was by measuring the water and the dust but noone knew for sure if this were dead viruses or live ones.

Therefore, I believe that it is important to produce novel diagnostics that are rapid and reliable and ubiquitous-thus they can protect the environment and the human and animal lives. Since there was no funding available for such innovative solutions, I believe that sharing these ideas and preliminary research results widely will influence science and engineering to look at these potential solutions too. The more widespread is this perspective the more people will be interested in exploring these ideas and the better quality of sensors and diagnostics will become available for the common good. This is what scientists should be doing anyway.

With respect to citing other work in the field, we've tried to address this in the updated manuscript as we did with the labeling of the figures and other omissions/errors.

Thank you for your support,

P. Gouma

Reviewer 3 Report

this paper presents an alternative, innovative and disruptive approach involving a live cell sensor microdevice that captures the viruses (and bacteria) from the air, gets infected by them, emitting signals for an early warning of the presence of pathogens. This to outline the processes and components required for living sensors to monitor the presence of pathogens in built environments and highlights the opportunity to use immune sentinels in cells of normal human skin to produce monitors of indoor air pollutants. producing a live cell microdevice can contribute a lot to the field of biosensing and biomedical applications as it can capture viruses and bacteria in air in more efficient and faster manner. I in general didn't find the system explained clearly and I didn't find the results and discussion enough to validate it is operation. moreover, better organizing for the cell cultures system and methodology of the system used for virus detection makes the paper more readable and less boring. there are some major revisions that need to be considered before accepting this manuscript:

1-Abstract: the motivation of this work to the SAR-COV-2 Pandemic is not clear, the author started by talking about this pandemic then talked breifly about his/her work, more emphasis about this work motivation and application needs to be explained by rewriting the abstract to be more related to the main system and its application 

2- How to design living cell sensors :

a) the author has used 16 self-citations from his/her work, which make the paper sounds like paraphrasing of his/her previous work without considering the other similar work done in literature, expanding the knowledge and connecting the current work to the previous work done by other is very important for deciding the originality and the value added by this work. 

b) not clear what does Figure 1 represents? is this a block diagram? for what? where are the tracking arrows? 

c) paragraph 3: starting from: " The key technical challenge is the response time..........." you need to cite enough references to explain the challenges in virus detection

d)you need to cite references for each technology mentioned in this part: The innovative technologies highlighted here include............................................

1.1. Cell Scaffolding:

a) paragraph 2 : no citations included, the complete paragraph needs to be written with citing enough references from literature   

b) Figure 2 : labels need to be added to each image, figure caption needs to be rewritten by adding the details of each image and the refernce/source of the image

1.2. Processing of the filtration membranes:

 a) Figure 3 : labels need to be added to each image, figure caption needs to be rewritten by adding the details of each image and the refernce/source of the image

 b) Figure 4: labels need to be added to each image, figure caption needs to be rewritten by adding the details of each image and the refernce/source of the image

2.2.3. D hybrid scaffolds and Pathogen Signatures:

  Figure 5 not 4 (as counted from previous figures): labels need to be added to each image, figure caption needs to be rewritten by adding the details of each image and the refernce/source of the image

3. Scalable Manufacturing and Validation of Air-Bio-Detectors:

a) figure 6 not 5 :  labels need to be added to each image, figure caption needs to be rewritten by adding the details of each image and the reference/source of the image

Author Response

Dear Sir/Madam

thank you for your thoughtful comments.

We have updated the manuscript to include more work in the literature on the cell electrospinning and the organs on a chip and the volatolomics approach and we have made all edits to respond to your comments.

Thank you for your support.

Yours,

P. Gouma

Round 2

Reviewer 3 Report

The authors have addressed all of my comments and suggestions, Thanks!

Author Response

-